# The Application of Bony Labyrinth Methods for Forensic Affinity Estimation

**DOI:** 10.3390/biology11071088

**Published:** 2022-07-21

**Authors:** Alexandra Uhl, Fotios Alexandros Karakostis, Katerina Harvati

**Affiliations:** 1Paleoanthropology, Institute of Archaeological Sciences and Senckenberg Center for Human Evolution and Palaeoenvironment, Eberhard Karls Universität Tübingen, 72070 Tübingen, Germany; alexandros.karakostis@uni-tuebingen.de (F.A.K.); katerina.harvati@ifu.uni-tuebingen.de (K.H.); 2DFG Centre for Advanced Studies “Words, Bones, Genes, Tools”, Eberhard Karls Universität Tübingen, 72070 Tübingen, Germany

**Keywords:** forensic science, forensic anthropology, population affinity, cranium, bony labyrinth, inner ear

## Abstract

**Simple Summary:**

When human remains are recovered, it is important to identify the individual for multiple reasons, including reuniting the individual with family members. In cases when human remains are incomplete, it can be challenging to identify the individual. This research found a highly successful method for identifying a person’s affinity by using measurements of the inner ear cavity. Since the inner ear is housed in a bony region that survives extremely well, the potential for identifying human remains is exciting. Researchers developed functions that can identify with 90.8% and higher accuracy between three different population samples using measurements from the bony labyrinth alone. These methods are non-destructive and quick to make, and plug into the functions developed in this research. This research points to potential for this method and calls for additional samples to be added to the data base to help with more identifications in the future.

**Abstract:**

Population affinity identification is important for reconstructing the biological profile of human skeletal remains. Most anthropological methods for predicting population affinity rely on complete crania or cranial parts. However, complete parts are frequently not found in forensic and bioarchaeological contexts. In contrast, the petrous portion of the cranium presents a unique rate of preservation in the field. Therefore, this study aimed to develop stepwise discriminant function formulae to determine population affinity using measurements on three-dimensional models of the human adult bony labyrinth. The sample utilised consisted of 30 German, 38 African Zulu, and 30 Oneota individuals. A total of four function equations were developed. The function involving all three populations presented an average accuracy of 90.8%. Mathematical equations were also derived to discriminate between Zulu and Germans (91.2%), Zulu and Oneota (95.5%), as well as Oneota and Germans (96.7%). These results indicate this new method of population affinity identification is highly successful, even with fragmentary remains.

## 1. Introduction

In forensic anthropology, assessing the population affinity of human skeletal remains comprises one of the primary steps for personal identification [1,2]. For this purpose, the cranium is the most frequently utilised anatomical part due to its increased heritability, which often allows the craniofacial morphology to preserve the underlying genetic structure of individuals [3].

The most widely applied methodology involves the macroscopic observation of cranial non-metric traits that tend to vary substantially across population affinities [1,4]. Nevertheless, the accuracy of this method can be largely affected by the skills and subjective experience of each researcher performing the analysis. Moreover, the accuracy and precision of this observational technique have rarely been verified using statistical analyses [4].

A series of other studies have developed craniometric methods for population affinity identification, usually based on linear discriminant function analysis (e.g., [5,6,7,8,9,10]. The precision of measuring cranial dimensions has been verified in most of these previous works. Some of them have reported excellent accuracy rates (e.g., [11]), which seem to exceed the accuracy reported recently for morphoscopic techniques [4].

However, further testing of these craniometric methods has demonstrated that their accuracy rates substantially drop when applied to different anthropological samples or modern forensic cases (e.g., [6,11,12]). This is often explained as the result of various factors of secular change, which extensively affect craniofacial variation across populations of distinct geographical and chronological backgrounds [13]. Consequently, craniometric methods for population affinity identification tend to be population specific and require systematic refinement [8,14]. Furthermore, most of the aforementioned methods require intact crania or cranial parts (e.g., vault or face), which are often missing from osteological contexts [2].

The petrous portion of the cranium is likely the most frequently found bone element at both crime scenes and archaeological sites [15], and it can even survive cremations unharmed [16]. However, until now, no methods have been developed to determine population affinity using the bony labyrinth of the adult human skull. This could be due to the particular anatomical position of the bony labyrinth in the skull that prevents its measurement using traditional craniometric techniques. 

In this framework, this study aims to propose a new and precise metric method for determining the population affinity of human skeletal remains using biometric data of documented adult bony labyrinths. The sample analysed involves individuals of European, African, and Native American origin. 

## 2. Materials and Methods

The samples used in this study consist of crania from 98 modern human adults of known sex and affinity from three geographic populations. Table 1 lists all specimens used. The Norris Farms Oneota sample dates to A.D. 1300, and was studied through permissions of the Illinois State Museum collections. The Zulu sample dates to the Early 20th century and was studied with the permission of the Richard Dart collection at the Univ. of the Witwatersrand, South Africa. The German sample dates to the 19th century in Baden-Württemberg and was studied with the permission of the Osteological collections at the E.K. Universität Tübingen. 

The virtual reconstruction of the crania was performed at each collection site. The Zulu sample was scanned at the Palaeosciences Centre Microfocus X-ray Computed Tomography (CT) Facility, University of the Witwatersrand with a Nikon Metrology XTH 225/320 LC at 100 kV, 105 µA, with an exposure of 500 ms per image, resolution between 60 and 120 µm, and a 1.8AI filter. The German sample was scanned at the Paleoanthropology High Resolution CT Laboratory, Eberhard Karls University Tübingen, using a Phoenix v|tome|x microCT scanner (General Electric) and 180 kV, 120 µA, 2500 images per scan with an exposure of 200 milliseconds (ms) per image and a resolution between 101 and 116 µm with a 0.1Cu filter. The slices pixel size was between 0.101 × 0.101 mm and 0.116 × 0.116 mm. The Norris Farms Oneota individuals were scanned with X-ray source energy settings at 150–180 kV and between 120 and 200 lA, with 2400 views and two–three samples per view. Slice thicknesses ranged from 0.0615 to 0.0781 mm, depending on specimen size. The field of view ranged from 63 to 82 mm, with pixel sizes ranging from 0.0615 to 0.0781 mm.

When available, both the right and left bony labyrinths were segmented utilising Avizo software (FEI Company, Hillsboro, OR, USA). Measurements of the three semi-circular canals and cochlea were made following Osipov et al. [17] by one of us (AU) on generated 3D virtual endocasts. Measurements included the width and height of each semi-circular canal and cochlea in millimetres (mm) (see Appendix A for a list of abbreviations and Uhl et al. [18] and [19] for more details). Following previous work (i.e., [17,18,20,21]), the mean of the left and right sides was used for analyses. The Oneota sample had fewer left temporal bones (*n* = 18) available for study than right temporal bones (*n* = 30); thus, for 12 of the Oneota, we did not use the mean and instead use the right-side measurements. Despite some individuals having only the right side, inspection of the accuracy was highly homogenous across all functions, for all samples, meaning the Onoeta had no considerable differences (see Results section). 

To assess intra-observer error, five individuals were measured three times each. The percent error for each measurement of the semi-circular canals and cochlea was below 5%, ranging from 0.76% to 3.35% (See Appendix A and [18]).

Using stepwise discriminant analysis, we created new cross-validated discriminant functions based on the samples studied here, which can be used to estimate the affinity of an unknown individual. All assumptions of discriminant function analysis were met [22,23]. The accuracy of each function was computed based on the discriminant scores of individuals both prior and post cross-validation (following a “leave one out classification” procedure) [23]. Furthermore, for each of the newly created discriminant functions, kappa statistics were applied to assess the level of agreement. To apply each newly created function to an individual of unknown affinity, the corresponding measurement is multiplied by the function coefficient for that variable. The resulting values are added to the function’s constant [23].

## 3. Results

Table 2 presents the summary descriptive statistics for the bony labyrinth measurements obtained. The aforementioned statistical tests demonstrated that all variables utilised have multivariate normality and do not present outliers. In all four discriminant function analyses performed, the Box’s M was shown to be non-significant, allowing for a variance–covariance matrix to be used. The repeatability analyses verified that there was no significant (*p*-value < 0.05) intra-observer error.

In three of the four discriminant analyses, based on the Wilk’s lambda statistics, the best-discriminating variable for population affinity determination was ChwM (Table 2). Contrastingly, in the analysis discriminating between Zulu and German individuals, the measurement PSChM was the most useful variable for the stepwise analysis. 

The function equations were built based on the unstandardised coefficients calculated (Table 3). For each function, the centroid of each population sample was computed. When classifying a newly found specimen, the process involves the multiplication of its measurements with their associated coefficients, followed by the addition of these quotients to the constant. Consequently, the sum (Y) is compared to the group centroids for each sample (Table 3), which were calculated based on the weighted group centroid values. The form of the equation is the following:Y = b_1_ × X_1_ + b_2_ × X_2_ + b_3_ × X_3_ +…+ b_i_ × X_i_ + a(1)
where: “b_1_ − b_i_” = regression coefficients (unstandardised coefficients), “X_1_ − X_i_” = the value of each variable, “a” = constant, and “i” = the number of predictor variables.

The accuracy rates of the four functions developed are demonstrated in Table 4, for both the original and the cross-validation samples. In the first two analyses, there are slight differences between the accuracy rates of the original sample and the cross-validated one (misclassification of a few individuals). In all four analyses, Kappa statistics revealed that there was “almost perfect” (k-value ranged between 0.814 and 0.933) agreement between the predicted and the actual population affinity of individuals (Table 4) [24]. 

In the first analysis (including all three population samples), the average accuracy rate was 90.8% for the original sample and 87.8% for the cross-validated one. All Oneota individuals were diagnosed correctly. Among the misclassified individuals, five were females and four were males. In the second analysis (including the German and the Zulu samples), the mean accuracy was 91.2% for the original sample and 89.7% for the one resulting after cross-validation. The incorrectly classified individuals involved three males and three females. The third equation, which discriminates between Zulu and Oneota individuals, presented an accuracy rate of 95.6% for both the original and the cross-validated samples. Only two females and one male were misclassified. Similarly, the function discriminating between Oneota and Germans correctly classified 96.7% of individuals. In both the original and the cross-validated samples, only two German females were incorrectly diagnosed. 

## 4. Discussion

The results of this study demonstrated that the human adult bony labyrinth can be used to assess population affinity with significant accuracy and precision. The slight drop in the accuracy rates in the cross-validated samples of the two first analyses did not result in substantial lowering of the predictive potential of the derived discriminant function equations (Table 4). The correct classification rate was slightly lower in the first analysis (involving three sample groups) than in the other three (each of them involving two sample groups). The highest rate was achieved by the function equation discriminating between Oneota and German individuals (96.7%). For all functions, a highly similar number of males and females was correctly assigned. 

In the literature, multiple studies have developed methods for determining population affinity using the human adult skull. One recent study utilised various statistic means to test the accuracy of using cranial non-metric traits for population affinity determination [4]. Based on that study, mean correct prediction rates reached 87.8%. 

Other studies have reported similarly high accuracy rates utilising craniometric dimensions (e.g., [5,6,8,10]. Steyn and Işcan [11] have reported an average accuracy of 98% in determining population affinity in a sample of South African black and white individuals, based on 13 standard cranial and 4 mandibular measurements. These authors have also developed mathematical equations for incomplete cranial remains (i.e., the vault and facial skeleton). However, standard craniometric measurements usually require complete crania or cranial parts, which are frequently absent or damaged in forensic and bioarchaeological contexts [2]. In these cases, the bony labyrinth, which presents excellent preservation rates [15] can be used for considerably accurate population affinity estimation. 

The accuracy of craniometric methods for population affinity identification is usually not consistent among populations of different geographical backgrounds [25]. This is possibly due to substantial craniometric variation across population groups. [11,13]. Similar remarks have been made concerning the metric methods that rely on postcranial measurements [25], which also present high accuracy rates (e.g., [26]. In this study, our method was considerably accurate for a population sample from an entirely different geographical and chronological background. This could suggest that the bony labyrinth preserves population history extremely well, most likely as a reflection of its surrounding temporal bone morphology, which has also been found to sustain signals of population history well [27,28]. Nevertheless, this result does not necessarily signify that population affinity is not a factor of variation in the human bony labyrinth. In the future, the applicability of our functions should be further verified using multiple population samples. Future research could focus on putting forth a method of population affinity estimation that would combine various standard craniometric distances with bony labyrinth dimensions.

Additional research is needed to enable the applicability of this approach to recent forensic cases, highlighting the important value of the present work as a basis for pursuing this goal in the future. Utilising the new technology of high-resolution microCT scans and processing of this large virtual data for precise measurements can allow future works to have larger sample sizes (such as the ones in this study), whereas former studies using this trait for forensic studies had smaller sample sizes, as they relied on direct linear measurements of bone.

## 5. Conclusions

Based on our sample, which consists of German, African (Zulu), and Oneota individuals, the human adult bony labyrinth can be used for accurate population affinity determination. When all three population samples were analysed together, the average correct classification rate was 90.8%. The mean accuracy rate of the function discriminating between Zulu and Germans was 91.2%, while the rate of the function involving Zulu and Oneota demonstrated an accuracy rate of 95.6%. The highest correct classification rate was presented by the mathematical equation discriminating between Oneota and Germans (96.6%). After applying the “leave-one-out classification” technique, the prediction accuracy was about the same. In all four analyses, a similar number of males and females were correctly diagnosed. This novel method of affinity estimation using the well-preserved bony labyrinth and non-destructive methods has high potential for identification of human remains, even if incomplete or in conditions where other methods are not possible.

## Figures and Tables

**Table 1 biology-11-01088-t001:** Sample demographics *.

Sample	Males	Females	Total
Norris Farms Oneota	16	14	30 *
Zulu	17	21	38
German	17	13	30
Total	50	48	98

* For the Norris Farms Oneota sample, both the left- and right-side bony labyrinth were available for 18 of 30 individuals. Analyses use mean values from the right and left variables, except for the 12 Oneota individuals, for which only right-side variables are used.

**Table 2 biology-11-01088-t002:** Stepwise discriminant function analysis of the bony labyrinth *.

Functions	Wilks’ Lambda Statistic	Exact F Statistic	d.f. 1	d.f. 2	Sig.
Function 1: All groups (Germans, Zulu, Oneota)					
ChwM	0.307	107,107	2	95,000	<0.001
PSChM	0.238	49,285	4	188,000	<0.001
PSCMR	0.215	35,896	6	186,000	<0.001
CwM	0.193	29,410	8	184,000	<0.001
CMR	0.170	25,988	10	182,000	<0.001
SLIM	0.156	23,014	12	180,000	<0.001
Function 2: Germans and Zulu					
PSChM	0.639	37,299	1	66,000	<0.001
SLIM	0.572	24,331	2	65,000	<0.001
ChM	0.517	19,924	3	64,000	<0.001
LSChw	0.461	18,392	4	63,000	<0.001
ChwM	0.433	16,212	5	62,000	<0.001
CwM	0.341	19,664	6	61,000	<0.001
Function 3: Zulu and Oneota					
ChwM	0.273	175,914	1	66,000	<0.001
ASCwM	0.232	107,452	2	65,000	<0.001
ASChw	0.206	82,353	3	64,000	<0.001
LSCMR	0.190	67,011	4	63,000	<0.001
(Removed ASCwM)			5		
Function 4: Germans and Oneota					
ChwM	0.244	179,220	1	58,000	<0.001 <0.001
PSChw	0.219	101,631	2	57,000	

* At each step, the variable that minimises the overall Wilks’ lambda is entered. Minimum partial F to enter is 3.84; maximum partial F to remove is 2.71. F values are all significant at *p* < 0.001 level.

**Table 3 biology-11-01088-t003:** Canonical discriminant function coefficients.

Functions	Unstandardised Coefficients ^1^	Structure Matrix ^2^	Standardised Coefficients	Group Centroids
Function 1: All groups (Germans, Zulu, Oneota)				
ChwM	−2025	−0.921	0.307	Germans: −1.704 Zulu: −0.461 Oneota: 2.288
PSChM	10,561	−0.291	−1.202	
PSCMR	4444	−0.198	1.227	
CwM	−19,100	0.529	0.2746	
CMR	0.014	0.111	−2.422	
SLIM	4849	−0.140	0.074	
(constant)	−7410			
Function 2: Germans and Zulu				
PSChM	1.031	0.541	0.611	Germans: 1.542 Zulu: −1.217
SLIM	−0.110	−0.158	−0.597	
ChM	29.012	0.385	−6.497	
LSChw	4.893	0.312	0.334	
ChwM	89.589	0.328	6.925	
CwM	29.012	0.090	7.753	
(constant)	−132.254			
Function 3: Zulu and Oneota				
ChwM	18.081	0.804	1.015	Zulu: 1.778 Oneota: −2.253
ASChw	14.660	0.202	0.552	
LSCMR	−2.514	−0.131	−0.554	
(constant)	−29.374			
Function 4: Germans and Oneota				
ChwM	11.804	0.931	0.859	Germans: 1.857 Oneota: −1.857
PSChw	6.590	0.537	0.372	
(constant)	−21.793			

^1^ Unstandardised canonical discriminant functions evaluated at group means. ^2^ Pooled within-groups correlations between discriminating variables and standardised canonical discriminant functions.

**Table 4 biology-11-01088-t004:** Accuracy of classification results of the original and cross-validated ^1^ samples.

Functions	Predicted Group Membership						
	Germans		Zulu		Oneota		Total Average (%)
	N	%	N	%	N	%	
Function 1: All groups (Germans, Zulu, Oneota)							
Original	25/30	83.3	34/38	89.5	30/30	100.0	90.8
Cross-validated	24/30	80.0	34/38	89.5	28/30	93.3	87.8
Function 2: Germans and Zulu							
Original	27/30	90	35/38	92.1			91.2
Cross-validated	26/30	86.7	35/38	92.1			89.7
Function 3: Zulu and Oneota							
Original			36/38	94.7	29/30	96.7	95.6
Cross-validated			36/38	94.7	29/30	96.7	95.6
Function 4: Germans and Oneota							
Original	28/30	93.3			30/30	100.0	96.7
Cross-validated	28/30	93.3			30/30	100.0	96.7

^1^ Cross-validation is performed only for those cases in the analysis. In cross-validation, each case is classified by the functions derived from all cases other than that case.

## Data Availability

Researchers interested in accessing databases may inquire with the corresponding author who can direct researchers to the appropriate collection’s managers for permission inquiries. Data collected will not be shared without collection managers’ permission.

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
