# Peer review of "The Application of Bony Labyrinth Methods for Forensic Affinity Estimation"

_biology, 2022, doi:10.3390/biology11071088_

Round 1

Reviewer 1 Report

Whereas the subject can be interesting, the way it is treated encloses some shortcomings, as follows. The authors need to decide whether this is about bioarchaeology or forensics. Please be aware that population affinities are not part of the bioarchaeological analysis.  In what concerns the forensic field, I disagree that "it is vital". Please be also aware that the terminology has changed. If you read the most recent publications ancestry has been replaced by population affinities. Also, pay attention that ethnicity is not the same and the same applies to ethnic identification. Besides, it cannot be done based on " cranial parts".  what is meant by "complete parts" what do you mean by parts? 

Another major shortcoming is the number of individuals per group which is very low, precluding statistical significance. Moreover, although cross-validation was performed,  I would like to see how does it work in a forensic sample. But, above all, it is the origin of the samples which is problematic: past samples are not adequate to derive forensic assumptions. It cannot be done.  

Regarding the state of the art of the metric and non-metric methods used in forensic anthropology, I am afraid there are some misconceptions and flaws. Non-metric methods do not depend on the skills of the observer anymore and there are statistical procedures like OSSA. 

I don t know methods that can be applied to the vault. Please specify.

In terms of references, besides the recent ones on population affinities (several) I missed others such as Rodrigues et al, 2020 (Med Sci Law).

Finally, the idea that a set of measurements on the inner ear is enough for population ancestry is too simplistic. I don't think the inner ear can tell all story. Is it exciting as it is stated? I disagree. Promissing? eventually, but not with this sample

Author Response

Whereas the subject can be interesting, the way it is treated encloses some shortcomings, as follows. The authors need to decide whether this is about bioarchaeology or forensics. Please be aware that population affinities are not part of the bioarchaeological analysis.  In what concerns the forensic field, I disagree that "it is vital". Please be also aware that the terminology has changed. If you read the most recent publications ancestry has been replaced by population affinities. Also, pay attention that ethnicity is not the same and the same applies to ethnic identification. Besides, it cannot be done based on " cranial parts".  what is meant by "complete parts" what do you mean by parts? 

Author Reply: We have changed the wording from “it is vital” to “important” (line 22). We have changed “ethnic” and “ancestry” to “population affinity” throughout the article. Cranial parts are used in population affinity estimation i.e. the nasal aperture shape…see for example please The Human Bone Manual by Tim White and Pieter Folkens, pg 400 the section on ancestry identification. “Parts” meaning portions or individual bones of the cranium, i.e. a partial temporal bone or a partial mandible may be recovered.

Another major shortcoming is the number of individuals per group which is very low, precluding statistical significance. Moreover, although cross-validation was performed,  I would like to see how does it work in a forensic sample. But, above all, it is the origin of the samples which is problematic: past samples are not adequate to derive forensic assumptions. It cannot be done.  

We fully understand the reviewer's concerns that the sample's bioarchaeological sample cannot be used to directly assess population affinity in a forensic medicolegal context involving living individuals of diverse populations from numerous geographical regions. Nevertheless, we strongly believe that the very high accuracy rates observed in our results (involving at least 30 well-preserved individuals per group) provide an excellent argument for expanding our methodology to forensic cases in the future (as suggested by the reviewer). We now introduced text (lines 231-233) in the Discussion clarifying that future research is needed to enable the applicability of this approach to forensic recent cases, highlighting the important value of the present work as a basis for pursuing this goal in the future.

We would also like to highlight that our sample sizes are similar to those of previous research on these structures. This is partly due to the requirements of our approach, which involves high-resolution microCT scans and processing of this large virtual data to obtain precise measurements of the bony labyrinth. We now mention in the text (233-237) the fact that other studies on this trait and using this technology are also often restricted to smaller sample sizes than do (for example) forensic studies relying on direct linear measurements on bone. We believe that our study is exemplary of the value of using newer technologies in anthropological sciences to increase the accuracy of our assessments on population affinity, albeit the suboptimal sample sizes.

Regarding the state of the art of the metric and non-metric methods used in forensic anthropology, I am afraid there are some misconceptions and flaws. Non-metric methods do not depend on the skills of the observer anymore and there are statistical procedures like OSSA. 

I don t know methods that can be applied to the vault. Please specify.

In terms of references, besides the recent ones on population affinities (several) I missed others such as Rodrigues et al, 2020 (Med Sci Law).

Finally, the idea that a set of measurements on the inner ear is enough for population ancestry is too simplistic. I don't think the inner ear can tell all story. Is it exciting as it is stated? I disagree. Promissing? eventually, but not with this sample

Please see The Human Bone Manual by Tim White and Pieter Folkens, pg 400  the section on ancestry identification for specific methods and techniques using the cranium and cranial parts. The authors feel the three samples represents geographically diverse humans and were selected carefully. The highly accurate results indicate the inner ear’s population specific morphology. The surrounding temporal bone morphology preserves population history well and so it is not surprising that the inner ear reflects this morphology (see ref 26 and 27).

Reviewer 2 Report

This manuscript entitled “The Application of Bony Labyrinth Methods for Forensic Affinity Estimation” provide a new method for estimating the personal affinity and ancestry by the bony labyrinth. These functions present a higher accuracy rate compared with the cranial non-metric traits. we think this study is very useful to response the poor preserved human skeletons. However, before the acceptance of this manuscript, we have some comments to the author to improve some aspects of this manuscript. The details as follows:

Q1: The author mentioned that “the petrous is the most frequently found bone element at both crime scenes and archaeological sites”. The sentence is not entirely correct, because the tooth is more frequently found bone element in archaeological sites.

Q2: Material. In paragraph3, “Measurements included the width and height of each semi- circular canal and cochlea in millimeters (mm) (see Uhl et al. [18] for more details)”. The raw data may be found in the reference. However, the current expression is unclear and tend to be confusing for readers. We suggest a need for more clarity in stating the source of raw data.

Q3: Conclusion. In line 217, the author pointed out that the "leave-one-out classification" technique was used for cross-validation in his study. Notably, you should specify the method of cross-validation in Method. Besides, why did the author select this method? Did you consider other approaches, such as K-fold Cross Validation or holdout cross validation?

Q4: Can you introduce the applicability of these discriminating functions? Did you test these discriminating functions in other populations?

Q5: What factors account for the differences in accuracy of these functions involving different populations?

Q6: The authors should unify the format of references.

For example, the reference 4 and 10:

Hefner JT, Spradley MK, Anderson B. Ancestry assessment using random forest modeling. Journal of Forensic Sciences. 2014 267May;59(3):583-9

Hefner, J.T. and Ousley, S.D., 2014. Statistical classification methods for estimating ancestry using morphoscopic traits. Journal 256 of Forensic Sciences, 59(4), pp.883-890.

Author Response

This manuscript entitled “The Application of Bony Labyrinth Methods for Forensic Affinity Estimation” provide a new method for estimating the personal affinity and ancestry by the bony labyrinth. These functions present a higher accuracy rate compared with the cranial non-metric traits. we think this study is very useful to response the poor preserved human skeletons. However, before the acceptance of this manuscript, we have some comments to the author to improve some aspects of this manuscript. The details as follows:

 Thank you

Q1: The author mentioned that “the petrous is the most frequently found bone element at both crime scenes and archaeological sites”. The sentence is not entirely correct, because the tooth is more frequently found bone element in archaeological sites.

 Teeth are indeed extremely well preserved and often found but not considered a bony element.

Q2: Material. In paragraph3, “Measurements included the width and height of each semi- circular canal and cochlea in millimeters (mm) (see Uhl et al. [18] for more details)”. The raw data may be found in the reference. However, the current expression is unclear and tend to be confusing for readers. We suggest a need for more clarity in stating the source of raw data.

 Thank you for the concern. The authors feel that the methodology is well established in the literature (see source 28, 18, and 17 please) and the explanation of “width and height of each semi-circular canal and cochlea” suffice to describe the simple but specific measurements made. The focus of this paper is not to reiterate established methods but rather to confirm their use in application to population affinity estimation. However, in order to assist with the abbreviation meanings, we have added a supplementary table (S1).

Q3: Conclusion. In line 217, the author pointed out that the "leave-one-out classification" technique was used for cross-validation in his study. Notably, you should specify the method of cross-validation in Method. Besides, why did the author select this method? Did you consider other approaches, such as K-fold Cross Validation or holdout cross validation?

 Cross-validation using the "leave-one-out classification" is a standard, very popular, and powerful process for evaluating the robusticity of the observed accuracy rates. We have used this in multiple bioarchaeological and forensic papers before (for example, studies by the authors of this work; e.g., Harvati et al. 2019 Nature; Karakostis et al. 2013 IJO; 2014 Anthrop Anz; 2018 Science Advances; and many others). Its power relies on the fact that it classified each and every case of the sample based on a formula developed on the basis (training) of all the other specimens in the sample (excluding the one tested). It is also the standard process recommended by the producer of the SPSS software we used (IBM; see also Field, 2013). It is the equivalent of "Jackknifing", which is also routinely used and recommended in the vast majority of applications. On this basis, we consider this process as an acceptable avenue for cross-validating our accuracy rates. We have the citations to the SPSS manual by Field et al. (2013) where cross-validation is mentioned in the methods section.

Q4: Can you introduce the applicability of these discriminating functions? Did you test these discriminating functions in other populations?

 These discriminating functions have not been tested on other populations. However, the results indicate how population-specific bony labyrinth morphology is. We would expect other populations to have similar levels of population-specificity and thus future research can pursue development of additional discriminant functions. The current functions could be applied to other samples, but the results would indicate only the closeness with the three samples used here (Zulu, Oneota, German). We look forward to the opportunity to expand using more samples. As explained in our response to another comment by the reviewer 1 above, we consider this study an excellent basis for future research on the applicability of our approach to recent medicolegal cases and contexts. We now discuss this limitation and our future plans in lines 231-233.

Q5: What factors account for the differences in accuracy of these functions involving different populations?

Such very small percentage differences among population groups (about 4 to 6% of differences) could simply be due to random chance and sampling within groups.

Q6: The authors should unify the format of references.

For example, the reference 4 and 10:

Hefner JT, Spradley MK, Anderson B. Ancestry assessment using random forest modeling. Journal of Forensic Sciences. 2014 267May;59(3):583-9

Hefner, J.T. and Ousley, S.D., 2014. Statistical classification methods for estimating ancestry using morphoscopic traits. Journal 256 of Forensic Sciences, 59(4), pp.883-890.

Thank you. We have corrected reference 4.

Reviewer 3 Report

The manuscript submitted by Uhl and collegues explores metric variation of the bony labyrinths and its application to ancestry estimation. It is a well-written and properly designed paper. I reccommend its publication after minor revision. I have a few comments that I would like to share with the authors.

1- Throughout the paper you refer to "ethnic groups", "ethnicity" and similar. Ethnicity is a cultural concept and refers to how the individuals perceive themselves in relation to other people; it does not necessarily correlates to ancestry or population (e.g. two ethnic groups can have the same ancestry and different ancenstry can be present in the same ethnic group). I don't think it is a proper term to be used in a biological paper. I would like it to be changed with "ancestry" or "population", that are biological concepts and can be explored using biological techniques.

2- In the introduction (line 63-64), you state "The petrous portion of the cranium is the most frequently found bone element at both 63 crime scenes and archaeological sites [15] while it can...". The data about differential preservation of petrous bone are not in paper [15]. Please provide a better reference or rephrase the sentence (eg. the petrous bone is LIKELY AMONG the most frequent... or something equivalent)

3- In the methods (line 104) "see Uhl et al. [18] for more details". If possible, please add a table with the description of the measurements used.
As it stands now, the acronyms in tabs 2 and 3 are nonsense to me.
Other possibilities are to explain the acronyms in the tables' caption; or adding a table in the supplementary information.

4- Discussion (lines 195-196). "The accuracy of craniometric methods for ethnicity identification is usually not consistent among populations of different geographical background [24]." I cannot find this statement in the cited publication. In any case, I think that having 3 populations very different one from the other (Oneota, Zulu, German) increases a lot the accuracy in discriminating the populations. Having "entirely different geographical backgrounds" helps in seeing differences. Likely, also for the petrous bone there are different accuracies among populations of different geographical backgrounds, but you can see it only when exploring ancestry inside different geographical groups (e.g. inside Europe, inside Africa, inside America). And I would expect that the highest accuracy would be in Africa, since it hosts most of the human variability. Maybe I didn't understand well what you meant. Could you explain better the last paragraph of the discussion, please?

Author Response

The manuscript submitted by Uhl and collegues explores metric variation of the bony labyrinths and its application to ancestry estimation. It is a well-written and properly designed paper. I reccommend its publication after minor revision. I have a few comments that I would like to share with the authors.

Thank you

1- Throughout the paper you refer to "ethnic groups", "ethnicity" and similar. Ethnicity is a cultural concept and refers to how the individuals perceive themselves in relation to other people; it does not necessarily correlates to ancestry or population (e.g. two ethnic groups can have the same ancestry and different ancenstry can be present in the same ethnic group). I don't think it is a proper term to be used in a biological paper. I would like it to be changed with "ancestry" or "population", that are biological concepts and can be explored using biological techniques.

Thank you, we have changed all instances of ethnic group to population affinity.

2- In the introduction (line 63-64), you state "The petrous portion of the cranium is the most frequently found bone element at both 63 crime scenes and archaeological sites [15] while it can...". The data about differential preservation of petrous bone are not in paper [15]. Please provide a better reference or rephrase the sentence (eg. the petrous bone is LIKELY AMONG the most frequent... or something equivalent)

We have adjusted the wording to better reflect Iscan (2005) please see line 71 for changes to now be “…is likely the most frequently…”

3- In the methods (line 104) "see Uhl et al. [18] for more details". If possible, please add a table with the description of the measurements used.
As it stands now, the acronyms in tabs 2 and 3 are nonsense to me.
Other possibilities are to explain the acronyms in the tables' caption; or adding a table in the supplementary information.

In order to assist with the abbreviation meanings we have added a supplementary table (S1). This table also directs readers to the original method description papers in the notes.

4- Discussion (lines 195-196). "The accuracy of craniometric methods for ethnicity identification is usually not consistent among populations of different geographical background [24]." I cannot find this statement in the cited publication. In any case, I think that having 3 populations very different one from the other (Oneota, Zulu, German) increases a lot the accuracy in discriminating the populations. Having "entirely different geographical backgrounds" helps in seeing differences. Likely, also for the petrous bone there are different accuracies among populations of different geographical backgrounds, but you can see it only when exploring ancestry inside different geographical groups (e.g. inside Europe, inside Africa, inside America). And I would expect that the highest accuracy would be in Africa, since it hosts most of the human variability. Maybe I didn't understand well what you meant. Could you explain better the last paragraph of the discussion, please?

The first sentence is in reference to the chapter on the assessment of ancestry and the concept of race and that much more work needs to be done to differentiate traditional craniometric methods that were developed on the concept of “race” and thus are prone to inaccuracy as race is a cultural concept not a biological one…new methodology that looks at samples from diverse geographical populations (like those used here, and those mentioned by the reviewer) can focus more specifically at biological differences in morphology that reflect geographic distance or population affinities.

Round 2

Reviewer 1 Report

The paper is now improved and it can be published as it is.

Reviewer 2 Report

After reading the new version of your manuscript, I think it deserves to be published. I noticed that you add information required and provide explanations for our problems. However, there are still some minor problems need to be noted.

Q1: In Table3, the text “Function 1: All groups (Germans, Zulu, Oneota)” should be typed in bold to be consistent with others.

Q2: Please provide the information of reference 29.